# Testosterone Biosynthesis from 4-Androstene-3,17-Dione Catalyzed via Bifunctional Ketoreductase

**Yi Wei** [1,2]**, Guangyao Mei** [3]**, Jinlin Zhao** [2]**, Shaoyang Zhang** [1,2]**, Wenping Qin** [2]**, Qing Sheng** [1,*]** and Zhongyi Yang** [2,*]

[1] College of Life Sciences and Medicine, Zhejiang Sci-Tech University, Hangzhou 310018, China; 202130802161@mails.zstu.edu.cn (Y.W.)

[2] School of Pharmaceutical Science, Taizhou University, 1139 Shifu Rd., Taizhou 318000, China

[3] Zhejiang Hongyuan Pharmaceutical Co., Ltd., Donghai Fourth Avenue, Linhai 317000, China; 13958539258@hongyuanpharm.com

[*] Correspondence: csheng@zstu.edu.cn (Q.S.); yangzhyi@126.com (Z.Y.)

**Abstract:** Testosterone (TS) is an important androgen drug and a precursor of steroid drug synthesis. Ketoreductase 2 (KR-2) (GenBank accession no. ABP64403.1) is observed to stereo-selectively catalyze the bioreduction of 4-androstene-3,17-dione (4-AD) to testosterone and contribute to the regeneration of NADH using isopropanol as a co-substrate. The Km value of KR-2 was 2.22 mmol/L with 4-AD, and the optimal pH was 6.5–7.0. The enzyme is stable and demonstrates relatively high-level enzyme activity at 40 °C. Acetone significantly inhibits this activity. This inhibition was overcome using an intermittent vacuum during the reaction process. Finally, the amount of TS reached 65.42 g/L after a 52 h reaction with 65.8 g/L 4-AD, 10% isopropanol, and 2 g/L β–NAD$^+$ at 40 °C, with a conversion rate of 98.73%. A total of 6.15 g of TS was obtained from 6.58 g of 4-AD after the reaction and purification; the HPLC purity was 99.82%, and the overall yield was 92.81%. This enzyme provides a promising route for the green biosynthesis of testosterone for industrial applications.

**Keywords:** testosterone; bifunctional ketoreductase; NADH regeneration; enzymatic synthesis





## 1. Introduction

Testosterone (TS) is an androgen drug with important physiological activity [1,2] and a key intermediate for synthesizing many high-value steroid drugs, such as methyltestosterone, testosterone heptate, and 6β-hydroxytestosterone [3].

In previous studies, testosterone has been chemically synthesized from indanone [4], olefinic nitrile oxide [5], and androstene-3,17-dione [6]. In the newer reports, the synthesis of testosterone was mostly carried out using starting intermediates with a cyclopentane–polyhydrophenanthrene structure, such as cholesterol, diosgenin, phytosterols, 4-dienosterone-3,17-dione (4-AD), etc. These intermediates can be biologically transformed into testosterone, and the process is relatively simple and cost-effective [7–10]. When 4-AD was used as the starting intermediate, testosterone was obtained after one-step enzymatic reduction, which is the preferred method for the industrial production of TS.

In mammals, TS is synthesized from 4-AD catalyzed by 17β-hydroxysteroid dehydrogenase (17β-HSD) [11]. In 2016, Shao et al. [12] co-expressed human 17β-HSD and Saccharomyces cerevisiae glucose-6-phosphate dehydrogenase (G6PDH) in *Pichia pastoris* GS115, and the co-expressed system produced 11.6 g/L of TS from 4-AD in 120 h. In 2021, Ding et al. [13] improved the activity of 17β-HSD using a rational design, and the TS yield increased by 197% to 3.98 g/L.

The 17β-HSD from fungi and bacteria has been reported for the enzymatic reduction in 4-AD to TS. In 2017, Fernandez-Cabezon et al. [14,15] cloned the genes encoding 17β-HSDs from the bacterium *Comamonas testosteroni* and the fungus *Cochliobolus lunatus*, and the engineered strains of *Mycobacterium* smegmatis produced high yields of TS from sterols or androst-4-ene-3,17-dione (AD). In 2019, Govinda Guevara et al. [16] cloned 17β-HSD

from *Cochliobolus lunatus* into *Rhodococcus ruber* Chol-4, and TS was synthesized from 4-AD, with a molar conversion rate of 61% using glucose for co-factor regeneration. The 4-AD can be synthesized from phytosterols. In 2022, DN. Tekucheva et al. [17] reported the two-stage transformation of phytosterol by the actinobacteria *Mycolicibacterium neoaurum* VKM Ac-1815D and *Nocardioides simplex* VKM Ac-2033D, which were capable of oxidizing the phytosterol side chain and reducing androstenedione at C17, respectively. A total testosterone yield of 53% was obtained using 10 g/L of phytosterol.

The 17β-HSD belongs to the short-chain dehydrogenase (SDR) family [18,19]. In addition to 17β-HSD, a few other SDRs have been reported for the biotransformation of 4-AD to testosterone. Zhou et al. [20] reported 4-AD transformation via alcohol dehydrogenase from *Ralstonia* sp. (RasADH) and commercial glucose dehydrogenase (GDH). A TS space–time yield of 1.65 g/L/h was achieved at a load of 10 g/L 4-AD. In 2022, Su et al. [21] reported that the Prelog enzyme from Pseudomonas can reduce 4-AD to TS in diastereomeric excess. They co-expressed this Prelog enzyme and formate dehydrogenase in *Escherichia coli* BL21 (DE3), resulting in a testosterone yield of 28.8 g/L.

These studies utilized formate dehydrogenase (FDH) and GDH to regenerate the co-enzyme NADH. No reports on the use of secondary alcohol dehydrogenase for co-enzyme regeneration in TS biosynthesis have been published. During the biosynthesis of dehydroepiandrosterone (DHEA) using 5-androstene-3,17-dione (5-AD), we unexpectedly discovered a secondary alcohol reductase that can reduce the 17 carbonyl groups of 5-AD to hydroxyl groups, and TS was found to be an impurity in the reaction mixture (Figure 1a). This secondary alcohol dehydrogenase (KR-2) belongs to ketoreductase and was formerly used as a bifunctional enzyme in the synthesis of the antiviral drug atazanavir [22] (Figure 1b). It catalyzed the reduction in the intermediate (3S)-1-chloro-2-oxo-3-(N-tert-butyloxycarbonyl)-4-phenylbutane (α-chloroketone) to (2R,3S)-1-chloro-2-hydroxy-3-(N-tert-butyloxycarbonyl)-4-phenylbutane (α-chlorohydrin) and also performed NADH regeneration by dehydrogenating isopropanol to acetone.

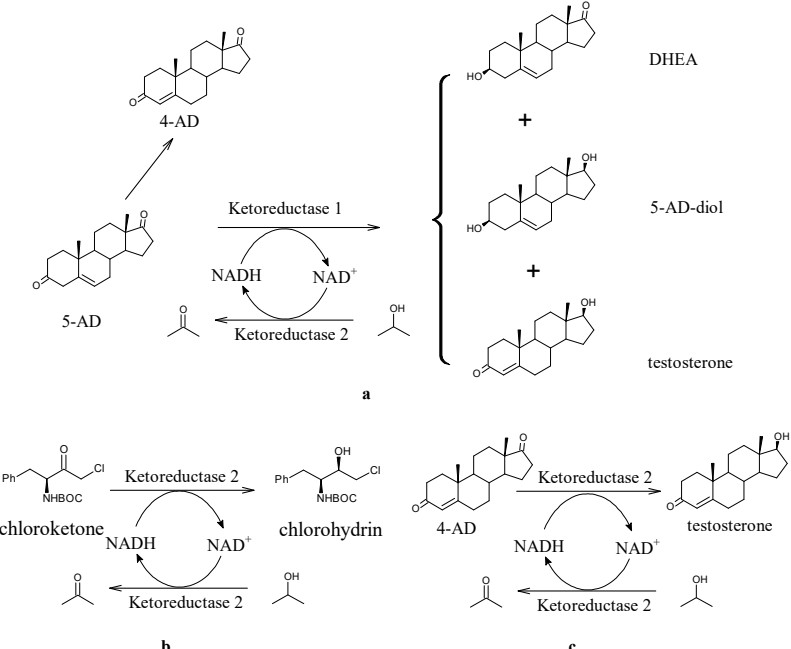

**Figure 1.** Biosynthetic manufacture of testosterone. (**a**) Enzymatic synthesis of dehydroepiandrosterone (DHEA) with ketoreductase. Ketoreductase 1 (KR-1): the *E. coli* BL21 (DE3)/pET-TZU21 strain was saved in the Enzyme Engineering Laboratory of Taizhou University (Taizhou, China). Ketoreductase 2 (KR-2): KR-2 is also a secondary alcohol dehydrogenase; the *E.coli* BL21 (DE3)/pET-TZU5 strain was saved in the Enzyme Engineering Laboratory of Taizhou University (Taizhou, China). (**b**) Enzymatic synthesis of chlorohydrin, an intermediate of atazanavir with ketoreductase 2. (**c**) Enzymatic synthesis of testosterone (TS) with ketoreductase 2.

Here, we performed TS biosynthesis from 4-AD with ketoreductase. It is interesting that TS was formed as the sole product when only ketoreductase 2 (KR-2) was used in the reaction, with 4-AD as the starting compound (Figure 1c). In this study, the TS concentration reached 65.42 g/L from 4-AD using KR-2, which is currently the highest reported level.

## 2. Materials and Methods

### 2.1. Chemicals and Media

The 4-AD was purchased from Goto Biopharm (Shiyan, China). The $\beta$–$NAD^+$ oxidized form (free acid, $NAD^+$) was from Sangon Biotech Co., Ltd., Shanghai, China. $KH_2PO_4$ and $K_2HPO_4$ were purchased from Aladdin Reagent Co., Ltd. (Shanghai, China). Tryptone and yeast extracts were purchased from Angel Yeast Co., Ltd. (Yichang, China). All the other reagents used in this study were of analytical grade.

### 2.2. Cultivation

The GenBank accession no. for KR-2 was ABP64403.1. The recombinant bacterium *Escherichia coli* BL21 (DE3)/pET-TZU5 cells were constructed and preserved at the Enzyme Engineering Laboratory of Taizhou University (Taizhou, China). Fermentation was performed in a 5 L bioreactor (Biotech-3BG, Shanghai BaoXing Bio-Engineering Equipment Co., Ltd., Shanghai, China). Low-temperature induction with $\alpha$-lactose was adopted to achieve a high protein expression level [23,24]. The *E. coli* BL21(DE3)/pTZU-5 strain was grown overnight at 37 °C at 220 rpm in 5 mL Luria broth (LB) with 100 μg/mL ampicillin. The overnight meat soup was inoculated in 100 mL of TB (2.4% yeast extract, 1.2% tryptone, 0.4% glycerol, 17 mmol/L $KH_2PO_4$, and 72 mmol/L $K_2HPO_4$) with 100 μg/mL ampicillin in disposable baffled flasks and grown at 37 °C at 220 rpm for approximately 4–5 h until the cells reached $OD_{600}$ 0.6–0.8. This preculture was used to seed 3.5 L TB medium in a 5 L fermenter. The bioreactor was run at a controlled temperature of 37 °C with >25% dissolved oxygen, and the aeration rate was maintained at 3.0 vvm. The culture was performed until the cell population reached $OD_{600}$. Subsequently, the culture temperature was reduced to 25 °C, and 150 mL of 20% (*w/v*) $\alpha$-lactose solution was added. Agitation cascade was employed to maintain the dissolved oxygen (DO) at >25%. Approximately 15 h of fermentation was required until the cell population reached $OD_{600}$ stability.

### 2.3. Enzyme Activity Assay

For the KR-2 activity assay, a 25 μL crude enzyme sample was mixed with a 0.975 mL reaction substrate (pH 7.0 50 mmol/L $KH_2PO_4$, 50 mmol/L $K_2HPO_4$, 0.1 mL isopropanol, 2 g/L 4-AD, and 0.5 g/L $NAD^+$) at 40 °C via shaking at 1100 rpm for 20 min in a metal bath. The sample was diluted 2–5-fold with a diluent (acetonitrile–water containing 0.1% phosphoric acid), centrifuged at $10,000\times g$ for 2 min, and analyzed using HPLC.

The activity of KR-2 was determined based on testosterone production. One unit of KR-2 is defined as the amount of enzyme required to catalyze the production of 1 μmol of testosterone per minute at 40 °C. The assay result was linear when the specific activity level of the sample was lower than 6.0 U/mL. The samples were diluted to ensure the result was lower than 6.0 U/mL after dilution.

### 2.4. Biosynthesis and Isolation of Testosterone

The optimization of the reaction conditions was performed in a 50 mL centrifuge tube; 2.0 mL of enzyme solution was mixed with 8 mL of reaction substrate mixture (containing 10 g/L of 4-AD, 1 mL of isopropanol, 1 g/L of $NAD^+$, and 7 mL of 50 mM pH 7.0 potassium phosphate buffer). The reaction was performed at 40 °C and 200 rpm in a water bath. The samples were diluted 10–100-fold and centrifuged at 10,000 rpm for 2 min for HPLC analysis.

A scaled-up reaction was conducted in a 250 mL four-necked flask. The reaction mixture contained 1.0–6.58 g of 4-AD, 10 mL of isopropanol, 100–200 mg of $NAD^+$, 20 mL of crude enzyme, and 70 mL of 50 mM pH 7.0 potassium phosphate buffer, with a total

volume of 100 mL. The reaction mixture was adjusted to pH 7.0 with 1 mol/L NaOH. The reaction was performed at 40 °C in a water bath.

To isolate and extract the testosterone, 300 mL of ethanol was added to the reaction solution. The mixture was then filtered, and the filtrate was concentrated under a vacuum to remove the solvent. The precipitate was recovered via filtration and recrystallized in ethanol. White powder was obtained and dried overnight at 50 °C to obtain the final product.

### 2.5. Analysis

The HPLC analysis of testosterone and 4-AD was performed as described by Han, with certain modifications [25]. The HPLC system used was the Agilent 1260 Infinity II HPLC system (Agilent Ltd., Santa Clara, CA, USA). A C18 (Supersil ODS2 5 μm 250 × 4.6 mm) reversed-phase analytical column was used at 50 °C. The mobile phase comprised 0.1% phosphoric acid and acetonitrile (40:60 *v/v*). The flow rate was 1 mL/min, and the detection wavelength was 240 nm. The retention times for testosterone and 4-AD were 5.697 min and 6.790 min, respectively.

To determine the optical rotation, the product (0.1 g) or testosterone standard was dissolved in 25 mL of ethanol, and the volume was increased to 100 mL with ethanol. The solution was loaded into a 20 cm long test tube, and optical rotation α was determined at 20 °C (589.44 nm). Nuclear magnetic resonance (NMR) analysis was performed at 101 MHz using chloroform [26].

## 3. Results and Discussion

### 3.1. Fermentation and Protein Expression

The fermentation of engineered *E. coli* BL21 (DE3)/pET-TZU-5 in a 5 L fermenter occurred similarly to a previously published procedure [27,28]. The enzyme activity is tightly coupled with cell growth (Figure 2a); both increased quickly at approximately 3–8 h and then slowly from 8 to 13 h. At the end of fermentation (13 h), the enzyme activity level was 1121 U/L, and the target protein accounted for approximately 24.5% of the total protein content. SDS-PAGE analysis showed that approximately 74% of the target protein was in the supernatant (Figure 2b).

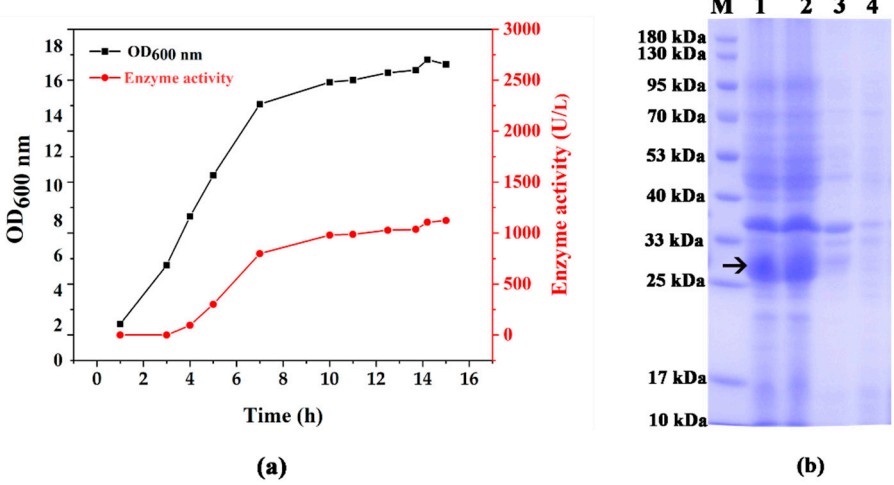

**Figure 2.** Fermentation process and protein expression. (**a**) Time course of low-temperature fermentation of KR-2 and changes in enzyme activity. (**b**) SDS-PAGE analysis of KR-2. Arrow pointing indicates target protein size. Lane M: molecular weight markers; lane 1: cell lysate with induction for 13 h at 25 °C. Lane 2: supernatant of cell lysate with induction for 13 h at 25 °C. Lane 3: sediments of cell lysate with induction for 5 h at 25 °C. Lane 4: cell lysate without induction.

### 3.2. Enzymatic Properties

KR-2 was very efficient in catalyzing α-chloroketone to α-chlorohydrin [22], but 4-AD has not been used as a substrate for KR-2 yet. Compared to α-chloroketone, the pH range

of the enzyme became more acidic when the substrate was 4-AD. The optimal pH was 6.5–7.0, and approximately 70% of enzyme activity was retained at a pH range of 6.0–9.0 (Figure 3a).

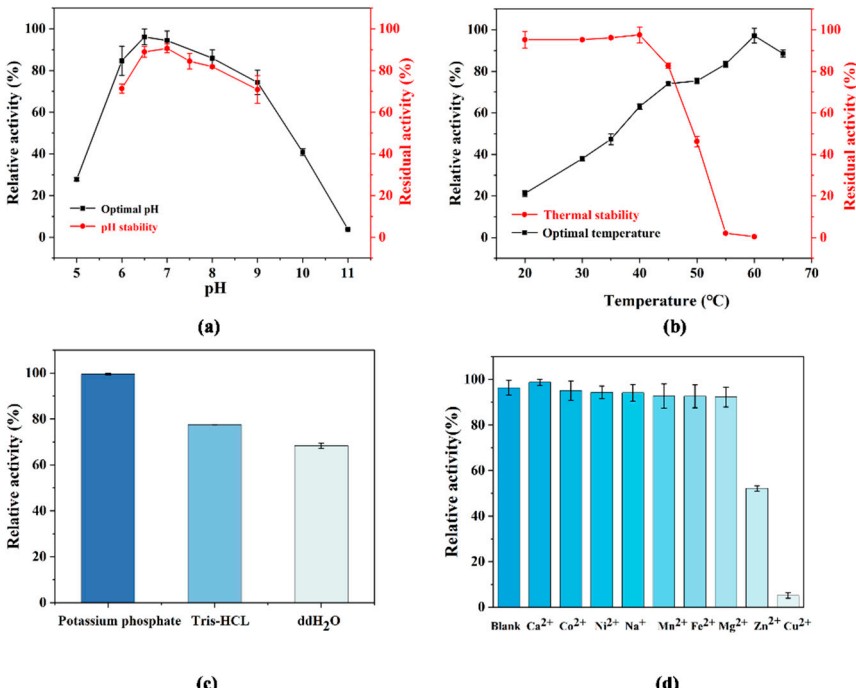

**Figure 3.** Enzymatic properties of KR-2. (**a**) Optimal pH and pH stability. The optimal pH was determined at 40 °C and within a pH range of 5.0–11.0. The residual activities were measured after incubating the enzyme at 40 °C for 2 h over a pH range of 6.0–9.0. (**b**) Optimal temperature and thermal stability. The optimal temperature was determined at pH 7.0 and within a temperature range of 20–70 °C. The residual activities were measured after incubating the enzyme at pH 7.0 for 2 h over a temperature range of 20–60 °C. (**c**) Effect of various buffers on enzyme activity. The enzyme activities were measured in different buffers (containing a 50 mM pH 7.0 potassium phosphate buffer, 50 mM pH 7.0 Tris-HCL, and ddH$_2$O). (**d**) Effect of various metal ions on enzyme activity. Reaction conditions: the 0.025 mL crude enzyme sample (supernatant of cell lysate) was mixed with the 0.975 mL reaction substrate (0.875 mL of 50 mM pH 7.0 Tris-HCL, 0.1 mL of isopropanol, 2 g/L of 4-AD, 0.5 g/L of NAD$^+$, and 1 mM of different metal ions) for reactions at 40 °C that were shaken at 1100 rpm for 20 min in a metal bath. The content of TS was determined using HPLC. All error bars represent SD for *n* = 3 independent experiments.

The effect of temperature on KR-2 was similar when the substrate was 4-AD. The enzyme activity level increased nearly linearly from 20 °C to 60 °C. At pH 7.0, the thermal stability of the enzyme seemed to be poor at temperatures higher than 40 °C. Only approximately 40% of the enzyme activity was retained after being stored for 2 h at 50 °C (Figure 3b).

Under pH 7.0 conditions, the activity of this enzyme in a phosphate buffer was approximately 25% higher than that in a Tris-HCl buffer and approximately 40% higher than without the buffer (Figure 3c).

With Tris-HCL as a buffer, we tested the effect of metal ions on the enzyme activity; most metal ions do not have a great effect on KR-2. In contrast, Cu$^{2+}$ Zn$^{2+}$ inhibited the enzyme activity. Only Cu$^{2+}$ exhibited strong inhibitory effects on the enzyme. The enzyme activity level decreased to 3% of that of the control in the presence of 1 mmol/L Cu$^{2+}$ (Figure 3d).

The Km and Vmax values of KR-2 were 2.22 mmol/L and 0.062 μmol/min with 4-AD, respectively.

### 3.3. Optimization of NADH Regeneration Conditions

The effects of isopropanol and NAD$^+$ on TS bioconversion were studied. Isopropanol is a substrate for NADH regeneration and a co-solvent used to improve the solubility of 4-AD. Among the four concentrations (5% $v/v$, 10% $v/v$, 20% $v/v$, and 30% $v/v$), the enzyme activity level was the highest in 10% isopropanol. Only 8% activity was retained when the concentration of isopropanol reached 30% $v/v$ (Figure 4a). The effect of isopropanol concentration on the reaction process was consistent with its effect on enzyme activity (Figure 4b). In the presence of 10% isopropanol, the reaction approached the endpoint within 10 h. With 5% isopropanol, it took approximately 25 h to reach more than 95% conversion, and the reaction slowed down after 9.5 h (approximately 74% conversion), likely because of the evaporation of isopropanol. Isopropanol at 20% and 30% seemed to reduce the enzyme stability, and the reaction became difficult after 10 h, with conversion rates of 80% and 10% at 25 h, respectively.

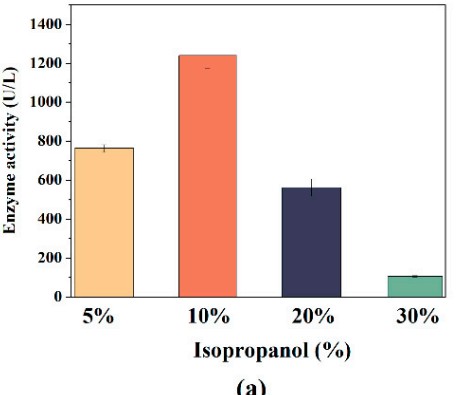 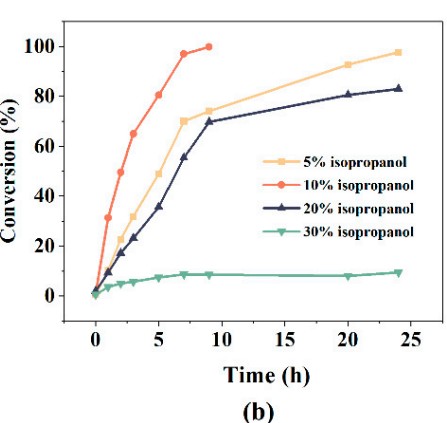

(a)          (b)

**Figure 4.** The effect of isopropanol on the synthesis of TS. (**a**) Effect of isopropanol on enzyme activity. Reaction conditions: 10 g/L of 4-AD, 1 g/L of NAD$^+$, 2 mL of crude enzyme, 0.5–3 mL of isopropanol, 5–7.5 mL of 50 mM pH 7.0 potassium phosphate buffer, reaction at 40 °C and pH 7.0, which were shaken at 200 rpm in a water bath. The content of TS was determined using HPLC. (**b**) Effect of isopropanol on the reaction process. Reaction conditions: the 0.025 mL crude enzyme sample (supernatant of cell lysate) was mixed with the 0.975 mL reaction substrate (0.925–0.675 mL of 50 mM potassium phosphate buffer, 0.05–0.3 mL of isopropanol, 2 g/L of 4-AD, 0.5 g/L of NAD$^+$, and pH 7.0) for reactions at 40 °C, which were shaken at 1100 rpm for 20 min in a metal bath. The *X*-axis represents the percentage of isopropanol in the total reaction volume ($v/v$). All error bars represent the SD for $n$ = 3 independent experiments.

NAD$^+$ is the central molecule of the regeneration system, and an appropriate NAD$^+$ concentration can ensure the smooth operation of the regeneration system and reduce costs. Under the conditions tested, the enzyme activity did not increase with an increasing NAD$^+$ concentration. However, the reaction process is tightly correlated with the NAD$^+$ concentration. The initial 1 h conversion rate of 0.25 g/L NAD$^+$ was only approximately 1/2 of the other three concentrations; it could be speculated that the Km value of this enzyme for NAD$^+$ was approximately 0.25 g/L.

As shown in Figure 1c, the main cause of the reaction was [4-AD][NADH]. The solubility of 4-AD was nearly stable, and the concentration of NADH during the reaction was roughly equal to the initial NAD$^+$ concentration; therefore, the initial NAD$^+$ concentration became the most important driving force for the reaction to quickly reach an equilibrium state. The times required for the four concentrations to reach the reaction endpoint of over 95% are >20 h, 20 h, 8 h, and 5 h, respectively (Figure 5). Considering the reaction time, conversion rate, and cost, 1 g/L of NAD$^+$ was selected for the next experiments.

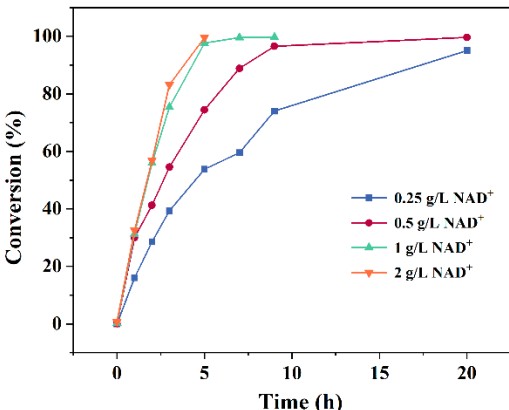

**Figure 5.** Effect of NAD⁺ on the reaction process. Reaction conditions: 10 g/L of 4-AD, 0.25–2 g/L NAD⁺, 2 mL of crude enzyme, 1 mL of isopropanol, 7 mL of 50 mM pH 7.0 potassium phosphate buffer, reaction at 40 °C and pH 7.0, which were shaken at 200 rpm in a water bath. The content of TS was determined using HPLC.

*3.4. Production of TS*

The production of TS was carried out at a 100 mL scale with a pH 7.0, 50 mmol/L potassium phosphate, 10% isopropanol, 1 g/L NAD⁺, and a 20 mL crude KR-2 solution at 40 °C. When the 4AD loads were 10 and 35 g/L, the reaction ended at 9 and 24 h, with final product concentrations of 9.69 g/L and 34.58 g/L and conversion rates of 99.29% and 98.12%, respectively (Figure 6a). However, when the 4-AD concentration was increased to 50 g/L, the TS concentration reached 34.81 g/L at 24 h. However, the extension of the reaction time and the addition of isopropanol and the enzyme only slightly improved the conversion. Theoretically, 7.10 g/L of acetone was produced in this reaction. Other experiments were then carried out to determine the effect of acetone on enzyme activity. The enzyme is susceptible to acetone, and its activity level dropped to 83% when the concentration of acetone was only 2 g/L. The enzyme activity level decreased more slowly with the addition of more acetone (Figure 6b).

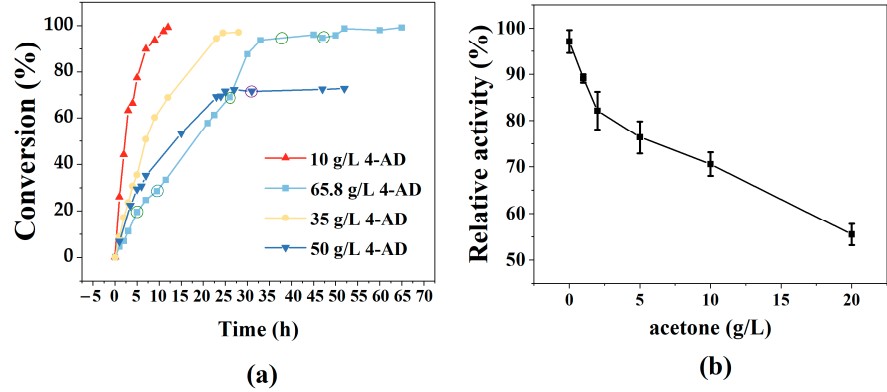

(a)　　　　　　　　　　　　　　　　　　　　(b)

**Figure 6.** Conversion process of TS and effect of acetone on the reaction process. (**a**) Conversion of testosterone to different substrate concentrations. The purple circle indicates the addition of isopropanol and the enzyme. The green circle represents the use of the vacuum. (**b**) Effect of acetone on the reaction process. Reaction conditions: the 0.025 mL crude enzyme sample (supernatant of cell lysate) was mixed with the 1.0 mL reaction substrate (pH 7.0 50 mmol/L $KH_2PO_4$, 50 mmol/L $K_2HPO_4$, 0.1 mL isopropanol, 2 g/L of 4-AD, and 0.5 g/L of NAD⁺) for the reaction at 40 °C via shaking at 1100 rpm for 20 min in a metal bath. All error bars represent the SD for *n* = 3 independent experiments.

To overcome the inhibition of acetone and further improve the efficiency of KR-2, acetone was removed using a vacuum in the reaction with 65.8 g/L 4-AD, 10% isopropanol, and 2 g/L NAD⁺ at 40 °C. The reaction solution was vacuumed for approximately 10 min at 5 h, 9.5 h, 26 h, 35 h, and 47 h, respectively, and then the reaction volume was replenished to 100 mL with water and isopropanol at a 9:1 ratio. The final TS reached 65.42 g/L after 52 h with a conversion rate of 98.73%.

To isolate TS from the reaction solution, three volumes of ethanol were added to dissolve TS and remove the proteins. After crystallization and recrystallization, unreacted 4-AD remained in the mother liquor, resulting in TS with a purity of over 99.5%. The total yields of the reaction and extraction exceeded 90% (Table 1).

**Table 1.** Results of the batch reaction on different substrate concentrations.

| Substrate Concentration (g/L) | Time (h) | Conversion (%) | Product Purity (%) | Product Yield (%) |
|---|---|---|---|---|
| 10 | 9 | 99.29 | 99.79 | 97.94 |
| 35 | 24 | 98.12 | 99.81 | 91.59 |
| 65.8 | 52 | 98.73 | 99.82 | 92.81 |

TS (6.15 g) was obtained from 6.58 g 4-AD after the reaction and purification of the last 100 mL of the reaction; the HPLC purity was 99.82%, and the overall yield was 92.81% (Figure 7a). The specific optical rotation $[\alpha]_D^{20}$ of this product and the TS standard were 102.33 and 102.26, respectively (c = 1, ethanol), which means that the configuration of the product was consistent with testosterone. The NMR spectrum of the product confirms its structure (Figure 7b,c).

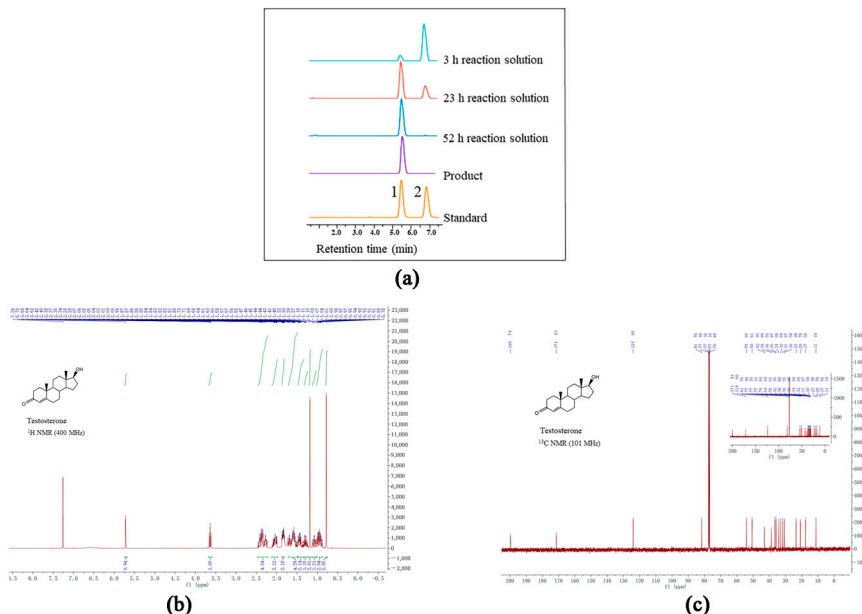

**Figure 7.** Liquid chromatograms of the conversion process and NMR spectra of the product. (**a**) Liquid chromatogram of the enzymatic synthesis of testosterone (1-testosterone; 2-4-AD). (**b**) ¹H NMR of product. (**c**) ¹³C NMR of product.

¹H NMR (400 MHz, Chloroform-*d*) δ 5.72 (s, 1H), 3.64 (t, *J* = 8.5 Hz, 1H), 2.46–2.23 (m, 4H), 2.12–1.98 (m, 2H), 1.9–1.76 (m, 2H), 1.76–1.48 (m, 4H), 1.48–1.31 (m, 2H), 1.31–1.23 (m, 1H), 1.18 (s, 3H), 1.13–1.02 (m, 1H), 1.02–0.86 (m, 3H), and 0.78 (s, 3H).

¹³C NMR (101 MHz, Chloroform-*d*) δ 199.74, 171.43, 124.00, 81.76, 77.36, 54.04, 50.61, 42.95, 38.80, 36.55, 35.87, 34.09, 32.94, 31.67, 30.58, 23.48, 20.78, 17.56, and 11.19.

Although the above process achieved a high concentration and high yield of TS, a lot of the enzyme was used in the reaction, which was relatively lengthy. The optimal temperature for this enzyme is 60 °C, but the stability of the enzyme sharply decreased when the temperature exceeded 40 °C, which hindered the catalytic efficiency. Immobilization can effectively improve the thermal stability of the enzymes and facilitate their recycling, which will be one of the future research directions of this project.

## 4. Conclusions

This study reports a method for efficiently synthesizing testosterone from 4-AD using one enzyme. Using this method, we obtained 65.42 g/L testosterone at a scale of 100 mL, which is currently the highest level reported.

The KR-2 used in this study belongs to the secondary alcohol dehydrogenase family of short-chain dehydrogenases. This enzyme catalyzes the reduction of substrate 4-AD to testosterone while catalyzing the regeneration of NADH with the co-substrate isopropanol. The enzyme activity level reached 1121 U/L in 5 L of the fermenter, and the crude enzyme solution was used to convert 4-AD. The inhibitory effect of acetone on the enzyme activity during co-enzyme regeneration could be overcome using the vacuum reaction method, which ensures the successful enzymatic synthesis of a high concentration of testosterone.

Several examples of secondary alcohol dehydrogenases used in co-enzyme regeneration are available. With further research, more secondary alcohol dehydrogenases could be discovered with a better co-enzyme regeneration ability and synthesis efficiency in catalyzing the synthesis of testosterone.

**Author Contributions:** Conceptualization, Z.Y. and G.M.; methodology, Y.W. and J.Z.; validation, S.Z.; formal analysis, W.Q.; investigation, Y.W., J.Z., S.Z. and W.Q.; resources, G.M., Q.S. and Z.Y.; data curation, Y.W. and J.Z.; writing—original draft preparation, Y.W.; writing—review and editing, Q.S. and Z.Y.; visualization, W.Q. and J.Z.; project administration, G.M. and Z.Y.; funding acquisition, Z.Y. All authors have read and agreed to the published version of the manuscript.

**Funding:** This research received no external funding.

**Institutional Review Board Statement:** Not applicable.

**Informed Consent Statement:** Not applicable.

**Data Availability Statement:** The data are contained within the manuscript.

**Conflicts of Interest:** Author Guangyao Mei was employed by the company Zhejiang Hongyuan Pharmaceutical Co., Ltd. The remaining authors declare that the research was conducted in the absence of any commercial or financial relationships that could be construed as a potential conflict of interest.

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
