# Peer review of "Testosterone Biosynthesis from 4-Androstene-3,17-Dione Catalyzed via Bifunctional Ketoreductase"

_fermentation, doi:10.3390/fermentation9120998_

Round 1

Reviewer 1 Report

Comments and Suggestions for Authors

The manuscript by Wei Y. et. al., “Testosterone biosynthesis from 4-AD catalyzed by bifunctional ketoreductase” reported that efficient synthesis of testosterone from 4-AD using an enzyme and achieved a concentration of 65.42 g/L of testosterone at a scale of 100 mL. However, the studies are fine and updated. There are few concerns to be addressed as follows:

Comments

1.     Section 2.3, please mention the exact amount of enzyme used in enzyme assay in terms of mg/mL or mg/L.

2.     The HPLC standard and reaction mixture chromatograph should be included by the authors.

3.     Lines 213-216, Authors should explain the allosteric effects of Ni2+ on enzyme activity.

4.     The discussion section is really poor and could be strengthened by comparative research.

5.     All figures must be increased in quality, and it is preferable to draw in color.

6.     The English language in the primary text needs to be improved.

7.     Abbreviations should be stated in the abstract and main text separately.

Comments on the Quality of English Language

Extensive editing of English language required

Reviewer 2 Report

Comments and Suggestions for Authors

Title: “by bifunctional 2 ketoreductase” This suggest two functions. Most reductases can reduce/oxidise the cofactor. Moreove, this not show that the same enzyme is used for cofactor regeneration.

Cofactor is not the same than coenzyme.

The use of the same enzyme for both reactions have advantages, as just one enzyme must be used, and problems, as it may be that the enzyme activity in both reavtions have very different optimal conditions (mainly pH).

Results offer many results from literature, this should be moved to introduction.

Figure 2. (a) Growth process of ketoreductase TZU-5’??? There are many sentences that require a better definition and writing

It may be interesting to have the curves in both, oxidation and reduction of the corresponding substrates, pH/activity .

They use free enzyme and find problems that can be related to lack of enzyme stability. Moreover, enzyme can not be easily reused. They can discuss than an evolution in the optimization of this process should be the enzyme immobilization, as this can prevent enzyme precipitation (likely in solvents), increasing enzyme stability (both thermal and in the presence of solvents), preventing enzyme subunit dissociation, and obviously enabling a simple enzyme reuse. There are multiple literature on these topics.

Comments on the Quality of English Language

Some sentenes seem to have the real menaning

Author Response

Dear Editor and reviewers,

  Thank you very much for your help in processing our manuscript (ID fermentation-2714588). We have carefully read the thoughtful comments, they are helpful for us to improve our manuscript. We have now completed the revision of our manuscript, the point to point responses to the reviewer’s comments are listed in the succeeding sheets. We hope that all these corrections and revisions would be satisfactory. Thanks a lot, again.

Sincerely,

Zhongyi Yang

Taizhou University

Shifu Rd, 1139

Taizhou, 318000

PR China

2023-11-17

  1. Title: by bifunctional 2 ketoreductase This suggest two functions. Most reductases can reduce/oxidise the cofactor. Moreove, this not show that the same enzyme is used for cofactor regeneration. Cofactor is not the same than coenzyme. The use of the same enzyme for both reactions have advantages, as just one enzyme must be used, and problems, as it may be that the enzyme activity in both reactions have very different optimal conditions (mainly pH).

Answer: Yes, here “bifunctional” means two functions of the same ketoreductase KR-2. In this paper, only one ketoreductase was used for the biosynthesis of testosterone, one function of this enzyme is the bio-reduction of 4-AD, the substrate, and the other function is the regeneration of NADH with isopropanol as substrate. Meanwhile, the other ketoreductase KR-1 mentioned in this paper is only capable of reducing 5-AD to DHEA, and cannot catalyze the regeneration of NADH with isopropanol as substrate. (page 2 Figure 1 and line 69 to 75)

  1. Results offer many results from literature, this should be moved to introduction.

Answer: Thank you for the advice. The paragraph from line 151-154 has been removed and integrated into introduction part (line 62 to 67). The paragraph from line 168 to 176 has been deleted, and some of content has been integrated in the next paragraphs as comparative results (page 5 line 162 to 171)

  1. Figure 2. (a) Growth process of ketoreductase TZU-5??? There are many sentences that require a better definition and writing

Answer: Thanks for pointing out the problem. The description of the Figure 2 (a) is incorrect. We have changed to “Time course of low-temperature fermentation of KR-2 and changes in enzyme activity” (page 4 line 156 to 157). Meanwhile, we redefine the name of the enzyme in this article (page 2 line 70 to 75). We have replaced TZU-5 with KR-2 throughout the article.

  1. It may be interesting to have the curves in both, oxidation and reduction of the corresponding substrates, pH/activity.

Answer: The pH was nearly stable in the reaction process. We didn’t analyze the concentration of isopropanol and acetone in the reaction solution. But since NAD+ concentration was the limiting factor in the reaction, the reduction speed of 4-AD and the oxidation speed of isopropanol should be equal, which meant that the apparent enzyme activity of the two functions was also equal.

  1. They use free enzyme and find problems that can be related to lack of enzyme stability. Moreover, enzyme can not be easily reused. They can discuss than an evolution in the optimization of this process should be the enzyme immobilization, as this can prevent enzyme precipitation (likely in solvents), increasing enzyme stability (both thermal and in the presence of solvents), preventing enzyme subunit dissociation, and obviously enabling a simple enzyme reuse. There are multiple literature on these topics.

Answer: Thanks for the advice, it’s a good research field for us in the future. We have added this in the discussion part. “Although the above process achieved high concentration and high yield of TS, the amount of enzyme used in the reaction was still large and the reaction time was relatively long. The optimal temperature for this enzyme is 60 ℃, but the stability of the enzyme sharply decreased when the temperature exceeded 40 ℃, which hindered the catalytic efficiency. Immobilization can effectively improve the thermal stability of enzymes and facilitate their recycling, which will be one of the future research directions of this project.” (page 8 line 287 to 292).